# Lung-Resident Mesenchymal Stem Cell Fates within Lung Cancer

**DOI:** 10.3390/cancers13184637

**Published:** 2021-09-16

**Authors:** Hanna Sentek, Diana Klein

**Affiliations:** Institute of Cell Biology (Cancer Research), University Hospital, University of Duisburg-Essen, 45122 Essen, Germany; hanna.sen@gmx.de

**Keywords:** mesenchymal stem cells, adventitia, vascular cells, pericytes, tumor stroma, cancer-associated fibroblasts, microenvironment, CAF, lung cancer, NSCLC

## Abstract

**Simple Summary:**

Lung cancer remains the leading cause of cancer-related deaths worldwide. Herein, the heterogeneous tumor stroma decisively impacts on tumor progression, therapy resistance, and, thus, poor clinical outcome. Among the numerous non-epithelial cells constructing the complex environment of lung carcinomas, mesenchymal stem cells (MSC) gained attraction being stromal precursor cells that could be recruited and ‘educated’ by lung cancer cells to adopt a tumor-associated MSC phenotype, serve as source for activated fibroblasts and presumably for vascular mural cells finally reinforcing tumor progression. Lung-resident MSCs should be considered as ‘local MSCs in stand by’ ready to be arranged within the cancer stroma.

**Abstract:**

Lung-resident mesenchymal stem cells (LR-MSCs) are non-hematopoietic multipotent stromal cells that predominately reside adventitial within lung blood vessels. Based on their self-renewal and differentiation properties, LR-MSCs turned out to be important regulators of normal lung homeostasis. LR-MSCs exert beneficial effects mainly by local secretion of various growth factors and cytokines that in turn foster pulmonary regeneration including suppression of inflammation. At the same time, MSCs derived from various tissues of origins represent the first choice of cells for cell-based therapeutic applications in clinical medicine. Particularly for various acute as well as chronic lung diseases, the therapeutic applications of exogenous MSCs were shown to mediate beneficial effects, hereby improving lung function and survival. In contrast, endogenous MSCs of normal lungs seem not to be sufficient for lung tissue protection or repair following a pathological trigger; LR-MSCs could even contribute to initiation and/or progression of lung diseases, particularly lung cancer because of their inherent tropism to migrate towards primary tumors and metastatic sites. However, the role of endogenous LR-MSCs to be multipotent tumor-associated (stromal) precursors remains to be unraveled. Here, we summarize the recent knowledge how ‘cancer-educated’ LR-MSCs impact on lung cancer with a focus on mesenchymal stem cell fates.

## 1. Introduction

Lung cancer is one of the most commonly diagnosed cancers worldwide and with remaining the leading cause of cancer-related deaths, the burden accounts for nearly 2 million deaths per year [1,2]. In terms of its diversity of phenotypes, lung cancer is heterogeneous and complex. Two main types of lung cancer can be distinguished, non-small and small cell lung cancer, with approximately 85% of patients suffering from the histological subtypes collectively known as non-small cell lung cancer (NSCLC), and further distinguishing its most common subtypes adenocarcinoma (AD) from squamous cell carcinoma (SCC) [3,4]. Although gained knowledge in understanding lung cancer biology and mechanisms of tumor progression have been achieved, and as a result, significant improvements in early detection and multimodal lung cancer therapy over the past two decades, overall cure and survival rates for NSCLC remain low, particularly in metastatic disease with varying five-year survival rates from 4–17% [1,3].

Beside the genetic alterations that determine oncogenic transformation of normal cells prior to sustained proliferation and apoptosis inhibition of malignant cells and, thus, tumor initiation and progression, the tumor’s stromal microenvironment contributes decisively to tumor heterogeneity as well as to complexity, and has become established as a leading player in carcinogenesis, particularly for lung cancer [5,6,7,8]. Cancer cells (‘the seed’) can impact on adjacent stromal cells (‘the soil’), finally resulting in cancer-educated stromal cells that in turn synergistically support tumor growth, progression, and potentially therapy resistance [8,9,10]. Activated fibroblasts (myofibroblasts and cancer-associated fibroblasts, CAF) and associated extracellular matrix, mesenchymal cells and vascular cells as well as infiltrating immune cells are central stromal cells, which define inter- and intratumoral heterogeneity. Lung SCC, for example, exhibit a higher intratumoral heterogeneity than lung AD whereas lung AD was found to have a higher metastatic rate than lung SCC [11,12]. Stroma-enriched tumors in NSCLC patients, in particular, usually have a worse prognosis and were associated with reduced disease-free periods compared to stroma-poor tumors [13]. Especially for advanced NSCLC, a more precise profiling of individual patients on stroma-cellular levels beyond the traditional pathological definitions would be highly desirable, because the identification of decisive tumor stroma cells with their respective origin, their relevant marker genes, and their specified intratumor cell–cell interactions could potentially serve as biomarkers and targets for individual therapeutic decisions for lung cancer patients at advanced stages [12].

Lung-resident mesenchymal stem cells (LR-MSCs), also designated as mesenchymal stromal cells (MSCs), are multipotent cells that are important orchestrators of normal lung homeostasis. Beside the potential to foster lung repair processes and, thus, tissue regeneration by differentiating into a variety of cell types replacing dysfunctional cells, LR-MSCs exert beneficial effects mainly by local secretion of various growth factors and cytokines that in turn regulate pulmonary tissue repair including inflammation. Accordingly, cell-based treatments using therapeutic applications of exogenous MSCs were shown to improve various lung diseases, including acute respiratory distress syndrome, pneumonia, chronic obstructive pulmonary diseases, and pulmonary fibrosis [14,15,16,17,18]. Even though safety and feasibility for MSC-based therapeutic approaches are figured out so far in pioneering clinical trials, the exact mechanisms of MSC actions remain elusive, while at the same time numerous preclinical studies already highlight possibilities to increase the therapeutic effectiveness of MSCs by modulating the respective secretome [16,19,20,21]. Beside an inherent tropism towards inflammatory and fibrotic sites, MSCs were shown to possess a natural tumor-trophic migration ability, strongly suggesting exogenous MSCs as promising pathotropic delivery vehicles when loaded with anti-cancer agents or engineered to express bioactive anti-cancer molecules [22,23,24]. So-called nano-engineered human MSCs, namely MSCs incorporated with drug-loaded nanoparticles (e.g., containing the anti-cancer drug paclitaxel or doxorubicin), or modified MSCs to improve their anti-tumor properties (e.g., following transduction to overexpress TNF-related apoptosis-inducting ligand (TRAIL), single interleukins, or interferons) were shown to reduce tumor growth by inducing apoptosis in tumor cells while sparing respective normal tissue cells [25,26,27,28]. MSCs alone [29], or more specifically MSC-derived factors [30], might even be sufficient for a beneficial targeting of lung cancer by increasing the expression of programmed death-ligand 1 in lung cancer cells, which then might account for an attenuation of the immune system against NSCLC cells [30].

Following lung damage however, LR-MSCs appear not to be sufficient for lung tissue protection, which is most likely based on the fact that LR-MSCs could decrease in numbers following a pathological trigger and/or become dysfunctional [16]. Moreover, their reparative functions have been shown to depend on the local (immune) microenvironment [31]. Thus, in contrast to the reported anti-tumorigenic influence of exogenous applied MSCs, MSCs of different origins were reported to impact on tumor initiation and progression (Figure 1). Due to their anatomic location, predominantly within the vascular wall of larger blood vessels, LR-MSCs could be the first line cells being in stand by for the interaction with tumor cells and/or tumor-secreted factors [16,32,33]. Rather than circulating and subsequently recruited bone marrow-derived MSCs, LR-MSCs could be mobilized from their niche towards lung tumors and be activated to differentiate into vascular mural cells, which in turn stabilize angiogenic blood vessels, finally resulting in stabilization and, thus, normalization of angiogenic tumor blood vessels. Likewise, LR-MSCs could even be the important source for activated and/or cancer-associated fibroblasts (CAF), thereby modulating the tumor microenvironment including the recruitment of circulating immune cells and, thus, impacting on cancer progression as well as therapy resistance.

However, the role of endogenous LR-MSCs to function as tumor-stromal precursors that can be mobilized, activated, and educated by lung-cancer cells remains unraveled. A better understanding of how endogenous lung MSCs contribute to the local microenvironment of lung cancer holds the potential to develop strategies to improve lung cancer therapy by targeting LR-MSCs derived stromal cells. Here, we review the current knowledge about LR-MSC fates within lung cancer and how LR-MSCs interact with NSCLC cells for promoting the development and invasiveness of lung tumors.

## 2. The Tumor-Promoting Action of Lung Cancer-Associated MSCs

Compared to normal (‘naïve’) LR-MSCs tumor-associated MSCs were endowed with tumor-supportive properties. Tumors, tumor-derived factors, associated hypoxia, and inflammation might induce LR-MSC activation and/or mobilization towards the tumor site. Within the complex tumor microenvironment, tumor-associated MSCs adopt then a phenotype that is distinct from those of normal tissue MSCs, finally fostering the tumor promoting nature of the tumor stroma [34,35]. Understanding these interactions, particularly the origin of intratumoral MSCs as well as the respectively altered phenotypes and functions would allow to manipulate the cross-talk between MSCs and NSCLC in order to beneficially target lung cancer progression.

Numerous in vitro and in vivo studies could show that MSCs derived from different origins impact on lung cancer. The tumor-promoting features of MSCs concerning lung cancer were highlighted when bone marrow-derived MSCs were shown to promote the survival of lung cancer cells via expansion of immunosuppressive immune cell subsets in the bone marrow, in the primary tumor and in the metastatic sites following cancer-education of recruited MSCs [36]. Likewise, MSC-secreted extracellular vesicles from human bone marrow-derived MSCs significantly increased lung cancer cell proliferation, survival, invasiveness, and epithelial-to-mesenchymal transition (EMT) as well as macrophage M2 polarization more efficiently under hypoxic and, thus, more tumor-like conditions in comparison to normal conditions without hypoxia [37]. More evidence of the paracrine nature between the MSC interactions with malignant cells within the lung was made when exosomes derived from hypoxic bone marrow-derived MSCs (containing the microRNAs miR-193a-3p, miR-210-3p, and miR-5100) fueled neighboring lung cancer cells, thereby promoting cancer cell invasion and EMT [38]. Further on, and indicating biomarker potential of MSC-specific features, plasma levels of these three plasma exosomal microRNAs even showed diagnostic accuracy to discriminate lung cancer patients with or without metastasis [38]. Addressing the mechanism of how MSCs directly affect the behavior of lung cancer cells was revealed when bone marrow-derived MSCs were shown to promote metastasis through activation of the tyrosine-protein kinase ABL-matrix metalloproteinase 9 (MMP9) signaling axis in a panel of lung cancer cells [39]. Increased levels of MMP9 in turn were linked to metastasis and reduced survival rates in lung AD patients, strongly suggesting that inhibiting the MSC-NSCLC crosstalk through inhibiting the ABL kinase-MMP9 signaling axis could be a potential target to reduce (MSC-induced) NSCLC metastasis [39].

Tumor-associated MSCs were further shown to interact with cancer stem cells and augment cancer stemness in human solid tumors [40]. Especially in desmoplastic cancers like NSCLC (but also in breast, colorectal, and pancreatic cancers), it is suggested that cancer cells alone and even in combination with applied cancer therapy educate recruited MSCs to acquire a tumor-associated MSC phenotype or induce MSC differentiation towards CAFs [40,41]. As a result, recruited MSCs contribute to the pro-tumorigenic microenvironment by secreting several pro-stemness chemokines such as interleukins (e.g., IL6), CC-chemokine ligands (e.g., CCL5), C-X-C motif chemokine ligands (e.g., CXCL12), and prostaglandins (e.g., PGE2). Thus, the heterotypic signaling between LR-MSCs and lung cancer cells is supposed to maintain a tumor-promoting ecosystem that could offer additional and new approaches for therapeutic interventions to improve lung cancer therapy. However, the definite origin of lung cancer-associated MSCs remains elusive, and especially whether indeed tissue-resident MSCs within lungs as ‘local MSCs in stand by’ or rather recruited MSCs (predominately from the bone marrow) arrange within the stroma of lung cancer.

That indeed LR-MSC significantly impact on NSCLC phenotype and progression was shown when conditioned media derived from MSCs either isolated from the patient’s normal lung tissue, from adjacent NSCLC tissues, or from the NSCLC metastatic bone marrow niche were used to stimulate NSCLC cells. Primary LR-MSCs isolated directly from NSCLC tissue promoted NSCLC cells’ proliferation, migration, autophagy, and EMT finally accounting for the expansion of malignant clones, an effect that was significantly pronounced compared to normal LR-MSCs [42]. In contrast, metastatic site MSCs rather facilitated the NSCLC cells invasive phenotype and re-seeding. Using more specifically microvesicles harvested from MSCs of primary and metastatic NSCLC niches confirmed the differential modulation of lung cancer cells by the MSC secretome [43,44]. A paired comparison of human lung cancer-associated MSCs and adjacent normal LR-MSCs further confirmed the differential tumor-modulating features of both lung MSC ‘variants’ [45]. The combined action of lung cancer cells within their altered pro-tumorigenic microenvironment induced a tumor-associated MSC phenotype in normal LR-MSCs that in turn promoted primary tumor cell dissemination. Accompanied gene expression profiling further identified a four-gene MSC signature comprising the genes GREM1 (Gremlin 1 DAN Family BMP Antagonist), LOXL2 (Lysyl Oxidase Like 2), ADAMTS12 (ADAM Metallopeptidase With Thrombospondin Type 1 Motif 12), and ITGA11 (Integrin Subunit Alpha 11) being induced in LR-MSCs following lung cancer education that functionally accounted for lung metastasis [45]. Thus, it could be demonstrated that LR-MSCs, or more specifically their neighboring lung cancer-associated MSCs, bear the potential for being a new hallmark that could be additionally used to improve the accuracy of prognosis prediction of lung cancer patients. Due to the heterogeneous nature of the tumor stroma, there is a hitherto lack of definitive tumor stroma-derived biomarkers and of particularly (tumor-associated) MSC-specific biomarkers. The identification of such would allow to specify and improve risk stratifications and prognosis predictions for lung cancer patients [33,38,45]. Accordingly, a newly established microenvironment mimetic in vitro cell culturing system for isolation and expansion of stromal progenitors from ex vivo cultured lung tumor specimen confirmed that lung cancer MSCs played a dominant role for the maintenance of cancer stem cell populations [46].

As the most originating sites of the different lung cancer types were correlated with the areas of distinct epithelial cells and their progenitor cells, even LR-MSCs could hypothetically lead to de-differentiation of lung stem cells, thereby initiating lung cancer; or at least differentiate into (divers) non-tumorigenic stromal cells that ultimately define the histological type of NSCLC [12,16,33]. Conformingly, MSC-related marker expressions (e.g., CD90, CD44, CD105, CD73 as well as LR-MSC specific HOX transcription factors) were associated with low pathologic stage in NSCLC and/or accounted for a reduced overall survival of respective patients [47,48,49,50]. Elevated MSC marker particularly in malignant lung epithelial cells could indicate that NSCLC may at least partially derive from LR-MSCs [33,47,48,49,50]. Conformingly, LR-MSCs were even found to be increased in numbers in NSCLC tissues [33]. Considering the fact that isolated LR-MSCs impacted on proliferation and survival of co-cultured NSCLC cells, it is more likely that LR-MSCs contribute as heterogeneous and non-tumorigenic cells to lung cancer, which affect the NSCLC histological type [33]. Supportive evidence was made when MSCs from lung cancer were shown to promote tumor metastasis and tumorigenesis both in vitro and in vivo while normal LR-MSCs as healthy MSC pair isolated adjacent tumor-free tissues failed to cause lymph node metastasis [51]. Lung cancer-associated MSCs here were shown to induce EMT in malignant lung cells and acquisition of stem-like traits [51]. Of note, compared to bone marrow-derived MSCs, lung cancer-associated MSCs were shown to produce multiple kinds of matrix metalloproteinases in much higher levels to modulate the extracellular matrix within the tumor stroma and then facilitate tumor migration [51].

Conclusively, these studies highlight that the mechanism of the endogenous LR-MSC-NSCLC cell interactions significantly impacts on NSCLC phenotype and progression and even more can be differentiated from bone marrow-derived MSC signaling. Thus, different targeting strategies would be required in accordance with niche/disease stage to inhibit lung cancer progression by manipulating the dialogue between LR-MSCs and NSCLC cells as innovative therapeutic approaches. Over all, the successful translation of potential lung cancer-associated biomarkers into the clinical setting is desired to augment lung cancer early detection, enabling a reduction of mortality and morbidity, especially in high-risk individuals [52,53]. MSC-derived biomolecules were even suggested to function as noninvasive biomarkers for lung cancer progression. Therefore, LR-MSC-related and respectively derived biomarkers from NSCLC educated LR-MSCs could guide the next-generation clinical studies.

## 3. LR-MSCs as Source of Activated Fibroblasts

Beside a more indirect contribution to cancer progression by interacting with respective tumor cells via paracrine signaling, MSCs could participate in tumor progression following differentiation into diverse cell types [54]. As the most abundant components of the tumor stoma, cancer-associated fibroblasts (CAFs) are key contributors to therapy resistance and poor long-term prognosis in cancer, particularly in NSCLC [55,56]. In general, CAFs were considered as tumor-associated fibroblasts that bear no reactivity for epithelial, endothelial, and leukocyte markers and lack the mutations found within associated cancer cells [56,57]. CAFs exert diverse tumor-promoting functions, including extracellular matrix secretion and remodeling, extensive bi-directional signaling communications with cancer cells and interactions with infiltrating leukocytes [57,58]. However, due to the lack of unique markers together with the heterogeneity in CAF phenotypes and functions, there is only limited success up to now to target these stromal cells for improving therapeutic benefits. The high heterogeneity found within CAF populations might be based on the numerous potential cellular predecessors of CAFs [56,57,59].

CAFs could result from the activation of normal tissue-resident fibroblasts or transdifferentiated from non-fibroblastic lineage such as epithelial and endothelial cells following EMT and endothelial-to-mesenchymal transition in response to stimulations by cancer cells and/or other associated stromal cells (e.g., with transforming growth factor (TGF), epidermal growth factor (EGF), platelet-derived growth factor (PDGF), and fibroblast growth factor (FGF) as key regulators) [57,58]. In addition, other recruited somatic cell types, e.g., adipocytes and circulating bone-marrow-derived mesenchymal cells could serve as sources for CAFs [34,57]. But -there is increasing evidence that CAFs possess many characteristics of MSCs, implicating these cells as a possible origin of CAFs [56]. Of note, naive MSC-originating tumor-associated MSCs, which decisively impact on cancer development, progression, and metastasis were shown to exhibit the potential to differentiate into CAFs [34,60].

Especially lung CAFs were shown to modulate NSCLC by promoting the self-renewal of respective lung cancer stem cells and enhancing drug resistance (e.g., to EGFR tyrosine kinase inhibitors) [61,62,63]. Subtypes of CAFs, namely podoplanin (a unique transmembrane glycoprotein receptor)-expressing CAFs within lung AD were found to be restricted in invasive rather than non-invasive AD, where CAFs promoted platelet aggregation and contributed to tumor formation and cancer cell invasiveness [64]. Likewise, fibroblast activating protein 1 (FAP1) expressing CAFs were associated with poor prognosis of NSCLC patients [65]. High desmoplastic CAFs in turn account for higher extracellular matrix remodeling rates that were shown to promote tumor invasion and growth in NSCLC patients [65]. Accordingly, high expression levels of ACTA2 (commonly known as alpha-smooth muscle actin or α-SMA), a frequently used marker for myofibroblasts/ CAFs were further associated with poor survival periods in NSCLC [66,67]. Thus, lung CAFs contribute to the malignant signature for advanced stage lung cancer [12]. 

In NSCLC, however, there are only a few studies yet reporting that the functional differences of lung CAFs were caused by the heterogeneity of origins. LR-MSCs could serve as precursors for lung CAFs. As tissue-resident stem cells, it is obvious that LR-MSCs could be activated by lung cancer cells and, subsequently, leading to the induction of differentiation into (activated) fibroblasts, finally yielding myofibroblasts or CAFs. Together with the suggestion that the heterogeneity of the CAFs origins is an important factor potentially determining CAF functions, unraveling the cross-talk between LR-MSCs and NSCLC could yield new promising targeting strategies to inhibit lung cancer progression and/or to improve outcomes in NSCLC. 

CAFs were already shown—with around 20% at least partially—to be derived from MSCs that were recruited from the bone marrow following tumor-dependent TGF and stromal cell derived factor (SDF) signaling [68]. Concerning tissue-resident MSC to CAF differentiation within cancer, it was shown that Gremlin 1 (an inhibitor in the TGF beta signaling pathway)-expressing MSCs were found to be increased within inflammation-induced gastric cancers, which served here as myofibroblasts precursors following SDF stimulation [69]. Although these findings were not established within lung cancer initiation and progression, the study revealed that MSC expansions were due to proliferation of tissue resident MSCs—at least at early time points of tumor initiation—since only small numbers of bone marrow-derived myofibroblasts were detected in respectively used chimeric transplantation studies [68,69].

One of the first indications that lung-cancer associated MSCs represent a progenitor reservoir for CAFs were established following a comparative molecular and functional analysis of human MSCs isolated from NSCLC and corresponding normal lung tissue [70]. In addition to the known MSC characteristics of both MSC ‘variants’, lung cancer-associated MSCs turned out to have higher proliferations rates while being less sensitive to chemotherapeutics as well as having increased expressions of genes involved in DNA repair, apoptosis, extracellular matrix synthesis, tissue remodeling, and angiogenesis [70]. Treatment with NSCLC-derived factors further on induced a CAF phenotype in both normal LR-MSCs and lung cancer-associated MSCs [70]. Thus, the idea was raised that effective stroma-targeted therapies in lung cancer should not only incorporate CAFs, but also their potential precursor cells, namely lung cancer-associated MSCs.

However, it remains elusive whether LR-MSCs as ‘migratory neighbors’ or other circulating (presumably bone marrow-derived) MSCs as ‘distant invaders’ [71] are recruited towards tumors and followed lung cancer cells instructions to become CAFs and/or tumor-associated MSCs themselves that manifest as the source of CAFs. Concerning the first assumption, although not derived from lung cancer studies, fate tracing using so-called pericyte-like MSCs residing closely associated to the pulmonary microvasculature revealed that these cells differentiated into (ACTA2-positive) myofibroblasts upon fibrosis development [72]. Emphasis on the fact that on site LR-MSCs and not bone marrow-derived MSCs differentiate into reactive fibroblasts was derived from an animal model of radiation-induced pneumopathy using bone marrow chimeras [73]. Following stimulation with the pro-fibrotic cytokines or direct lung MSC damage (e.g., upon thoracic radiations), endogenous LR-MSCs, and not those recruited from distant sites, promoted fibrotic remodeling by acquiring a pro-fibrotic myofibroblast phenotype [73,74,75]. 

Supportive evidence that lung tumor-derived MSCs potentially differentiate into CAFs in lung cancer was provided when respectively isolated MSCs from healthy and lung cancer tissues were compared [76]. Unlike lung cancer cells in here, isolated tumor-associated MSCs lacked respective malignant epithelial cell (EGFR) mutations but exhibited classical MSC characteristics. Additionally, tumor-derived MSCs expressed classical CAF markers as compared to normal LR-MSCs [76]. The adoption of a CAF-like phenotype was further proven in vitro following stimulations of isolated MSCs with tumor-derived factors, an observation that strongly suggested that tumor-associated MSCs differentiation towards CAF phenotypes in lung cancer is achieved by the lungs tumoral microenvironment [76]. Of note, lung cancer cells seemed to be the main instructor for inducing MSC differentiation towards CAFs. Using other MSC sources than the lung, namely adipose-derived MSCs, it could be shown that these cells exhibited an activated CAF-like phenotype following stimulation with NSCLC cell derived factors [77]. The MSC-derived CAFs in turn promoted lung cancer cell proliferation and invasion by elevating matrix metalloprotease expression as well as EMT [77].

Conclusively, MSCs significantly contribute to the intratumoral heterogeneity as central actors—beside CAFs—within the stromal compartment of lung tumors, strongly suggesting that MSC-targeted therapies could offer novel opportunities for enhancing the therapy response. Further cellular and associated molecular characterizations are needed, in order to unravel stroma-mediated NSCLC resistance mechanisms that could be potentially targeted by novel therapeutic strategies. Unfortunately, the lack of unique MSC, fibroblast and CAF makers, together with their highly overlapping marker panel (due to their shared mesenchymal origin), impedes gaining conclusive knowledge about the origin of the tumor stroma, particularly of specific cell types herein.

## 4. LR-MSCs as Mural Cell Precursors

The formation of a new vasculature is a central event in tumor development and progression, when angiogenic endothelial cells and/or endothelial progenitor cells form immature blood vessels that firstly lack coverage by stabilizing mural cells, and, thus, are considered being functional inferior [78,79,80]. Subsequently, association with and integration of pericytes and smooth muscle cells stabilize these immature vessels resulting in normalization of the tumor neo-vasculature [80,81,82]. Similar to the ‘CAF situation’, the origin of these cells is not finally clarified, but numerous evidence point towards the initial suggestion that, postnatally, these mural cells were generated by in situ differentiation from local mesenchymal stem or progenitor cells [83]. According to their adventitial location and, thus, their distribution throughout the body, vascular wall-resident MSCs were supposed to play a superior role as vascular mural cell progenitor cells [32,84,85,86]. As so, stem cell antigen-1-positive MSC-like cells within the vascular wall were identified to serve as local vascular smooth muscle progenitor cells [87,88]. Similarly, vascular wall-resident CD44-positive MSCs within the adult human vascular adventitia were shown to differentiate into pericytes and smooth muscle cells [89]. Therefore, it was suggested that vascular wall resident stem and progenitor cells, particularly endothelial progenitor cells and MSCs are the first line cells, which are available on the basis of their anatomic location as first point of contact for adjacent tumor cells [32,84,85,86].

Accordingly, although derived from a preclinical pulmonary fibrosis model, perivascular ABCG2 (ATP Binding Cassette Subfamily G Member 2)-positive LR-MSCs were identified as pericyte source that contributed to detrimental tissue remodeling in lung fibrosis [75]. An example for MSC to pericyte differentiation within the process of cancer progression was provided when glioblastoma-derived MSCs were shown to become activated following MSC-glioblastoma cells bi-directional signaling, finally resulting in MSC differentiation into pericytes that maintained tumor vascular structures, although the origin of glioma-derived MSCs remained elusive [90,91]. The angiogenic potential of human MSCs was further highlighted by respective expression profiles of proangiogenic factors leading to pericyte differentiation [92]. Mechanistically, recruited MSC to pericytes and/or smooth muscle cell differentiation as well as association with the nascent vasculature depends on several signaling networks, including the decisive angiogenic factors PDGF, VEGF (vascular endothelial growth factor) and TGF [93,94,95].

However, there is only little knowledge regarding the mechanisms of phenotype transformation from normal LR-MSCs to tumor-associated MSCs and subsequent differentiation into mural cells or the direct recruitment of LR-MSCs by tumors and subsequent pericyte and/or smooth muscle cell differentiation. Using Lewis lung xenograft tumors growing on bone-marrow chimeras of wildtype and Nestin-GFP transgenic mice, it could be shown that tissue-resident and the MSC marker Nestin expressing MSCs, presumably derived from the vascular wall, and not bone marrow-derived MSCs, were mobilized from their niche following MSC-tumor interactions. In turn, recruited MSCs contributed to vascular remodeling of the neovasculature by differentiating into pericytes as well as smooth muscle cells [96]. In contrast, pericytes differentiated from bone marrow-derived MSCs following chemokine (C-X-C motif) receptor 4 -dependent recruitment to lungs were shown to promote tumor recurrence via vasculogenesis after applying stereotactic body radiation therapy to Lewis lung carcinomas [97].

Over all, studies reporting the recruitment of endogenous MSCs, and particularly LR-MSCs, are rare. Lung cancer sites as well as associated treatment alterations produce numerous bioactive molecules and, thus, soluble mediators that in turn differentially influence LR-MSC fates as well as the recruitment of MSCs from other origin(s). Although lineage tracing in combination with preclinical lung disease models highlight that particularly LR-MSCs are important key players in lung injuries and especially in lung cancer, it will be necessary to specify the role of LR-MSCs within lung cancer including the development of suitable and noninvasive imaging strategies to monitor MSCs [16,33]. A combined targeting of vascular mural cells, particularly pericytes or their respective cells of origin, potentially LR-MSCs together with endothelial cells, may determine the success of anti-tumor therapies, as the prognosis of single anti-angiogenesis therapies remains poor [98,99,100].

## 5. The Challenges Regarding LR-MSCs

It has to be noted that the minimal criteria for defining the MSC phenotype, particularly plastic adherence and expression of known MSC markers (CD44, CD90, CD73, and CD105 while lacking the lineage markers CD45, CD34, CD14 or CD11b, CD79alpha or CD19 and HLA-DR) [101,102] turned out to be quite unspecific for phenotypic MSC characterizations as these cellular features were shared by different mesenchymal cells residing close by (Figure 2). Among the different pulmonary mesenchymal cells, fibroblasts (including lipo-, myo-, and adventitial fibroblasts), pericytes and smooth muscle cells, and mesothelial cells were commonly reported to share the highly similar mesenchymal phenotype, particularly the MSC-marker expressions with LR-MSCs, although their respective transcriptomic profiles differ [103,104]. Gaining knowledge of LR-MSCs, their associated niche(s), as well as their fate during lung cancer in contrast to other mesenchyme-derived cells close by, therefore, urgently requires the identification of additional cell type-specific markers and/or additional functional features.

LR-MSCs, for example, were shown to exhibit the stem cell feature known as the ‘side population phenotype’ that is based on the expression of ABCG2, a known multidrug transporter in cell membranes that enables various stem cells (and cancer cells) to efflux chemicals [108]. ABCG2-expressing MSCs derived from lungs were further shown to express the transcription factor Gli1 and the platelet-derived growth factor receptor beta in addition to classical MSC markers [109].

Until now, however, a combination of shared mesenchymal markers in combination with their respective lung localization is used to distinguish the different lung mesenchymal cells, particularly LR-MSCs. Numerous reports investigating LR-MSCs herein described respective cells as being localized perivascular and, thus, localized vessel-associated, although the precise MSCs niche(s) within lungs remained elusive [16]. Immunohistological in situ investigations then confirmed the vasculogenic zone of adult lung arteries as vascular stem cell niche for LR-MSCs. Together with the fact that LR-MSCs were phenotypically and functionally indistinguishable from vascular wall-resident MSCs (VW-MSCs) including a VW-MSC-specific HOX code, LR-MSC were suggested to be VW-MSCs [33]. In contrast to (adventitial) fibroblasts, which were defined as adherent cells of mesenchymal origin (while being non-endothelial, non-epithelial, and non-hematopoietic), but share the vascular wall localization within lungs, the differentiation potential as well as the colony-forming capacity were found to represent specific properties enabling to distinguish MSCs from fibroblasts [110,111,112].

Likewise, the different stem cell niches within lungs might impact on the LR-MSC phenotypes and functions (Figure 2). Within the alveolar (stem) cell niche, different mesenchyme-derived cell types and respective progenitor cells can be found that at least partially share MSC-like features [113,114]. As an example, the presence of (Wnt-responsive) mesenchymal cell populations expressing Axin2 and the platelet derived growth factor receptor alpha within lungs, designated as mesenchymal alveolar niche cells, was reported [113]. Based on the expression of leucine-rich repeat-containing G protein-coupled receptor-5 (Lgr5) and Lgr6, two additional mesenchymal populations were identified, Lgr6-expressing mesenchymal cells surrounding bronchiolar epithelia and being present in the alveolar space, and the mainly alveolar localized Lgr5-expressing mesenchyme cells [115]. The mesenchymal cells here seemed to be decisive for the region-specific crosstalk with adjacent airway stem cells in order to ensure proper lung tissue integrity [115]. However, it remains to be investigated whether these populations comprise indeed different subsets of mesenchymal progenitor cells, MSC-like cells, or maybe considered as MSCs.

Even similar to LR-MSCs, pericytes were shown to exhibit significant pro-metastatic features upon cancer progression [116]. Importantly, the localization of pericytes may be decisive for distinguishing them from LR-MSCs (Figure 2). Initially discovered as ‘Rouget cells’ in 1873, pericytes have become known as contractile vascular wall cells surrounding endothelial cells of the smallest blood vessels, the capillaries [117,118]. Lung pericytes, as defined at the level of the alveolar capillary bed being embedded within the capillary basement membrane of alveoli, were shown to express the classical MSC marker proteins [119,120]. This combines the currently accepted definition of mature pericytes being embedded within the vascular basement membrane in micro-vessels as visualized by of electron microscopy [121] with the observations of subendothelial pericyte-like cells that can be found in larger blood vessels [117,118]. The term pericyte, therefore, is frequently used in the literature to denote any microvascular (peri-) endothelial mesenchymal cell [117]. Thus, based on their anatomic location, pericytes cannot be found within the adventitia and/or perivascular. Upon removal from their endogenous niche, however, pericytes might acquire MSC-like features, namely a reported differentiation potential.

However, LR-MSCs further turned out to differ from the pericyte populations residing within lungs; the latter one expressing the integral membrane proteoglycan NG2. The monoclonal antibody 3G5-defined ganglioside antigen may further be used to distinguish pericytes within lungs, and potentially the lack of CD146 (known melanoma cell adhesion molecule) [119,120]. Again, one important aspect to identify and/or discriminate adult progenitor and stem cells from phenotypically similar cells are the capabilities of self-renewal and of multipotency. Using the transcription factor Tbx18 as selective pericyte marker in combination with lineage-tracing experiments (via an inducible Tbx18-CreERT2 expressions), it was successfully and convincingly shown that pericytes (and vascular smooth muscle cells) lack multipotency and maintained their identity during the course of life and, even in diverse pathological settings [122].

The precise identification of LR-MSCs, including the localization within lungs in combination with differential markers, adjacent to other mesenchymal lung cells would finally allow the improvement of drug interventions respectively targeting LR-MSC proliferation, migration and potentially activation within and out of their respective niche.

## 6. Conclusions

Based on their innate affinity to home to tumor tissues together with their known lung trapping after systemic infusion, therapeutically applied MSCs gained attraction for the targeted delivery of anti-cancer therapeutics aiming to improve current treatment modalities, particularly in NSCLC. In contrast, LR-MSCs in their native environment turned out to orchestrate the fate of tumor cells. LR-MSCs can be considered as key players in the initiation, malignance, and resistance of lung cancer. However, based on the reliance of MSC isolations of different origins, especially from the bone marrow, and subsequent MSC culture expansions, most of the studies investigating MSC recruitments, MSC-tumor interactions, and, thus, MSC fates within lung cancer are based on culture expanded MSCs. In contrast, there is a lack of studies addressing endogenous MSCs. As ‘migratory neighbors’ more likely than ‘distant invaders’, LR-MSCs are ready to join building up the lung tumor stroma: LR-MSCs could adopt a tumor-associated MSC phenotype and/or serve as source for activated fibroblasts and presumably for vascular mural cells following ‘instructions’ supplied by lung cancer cells and the associated pro-tumorigenic microenvironment, finally reinforcing tumor progression. A detailed understanding of signaling pathways controlling LR-MSC fates is urgently needed to unravel how endogenous MSCs foster lung cancer progression and how potential on-site manipulations of the LR-MSC/lung cancer cell interactions could be used for therapeutic benefits.

## Figures and Tables

**Figure 1 cancers-13-04637-f001:**
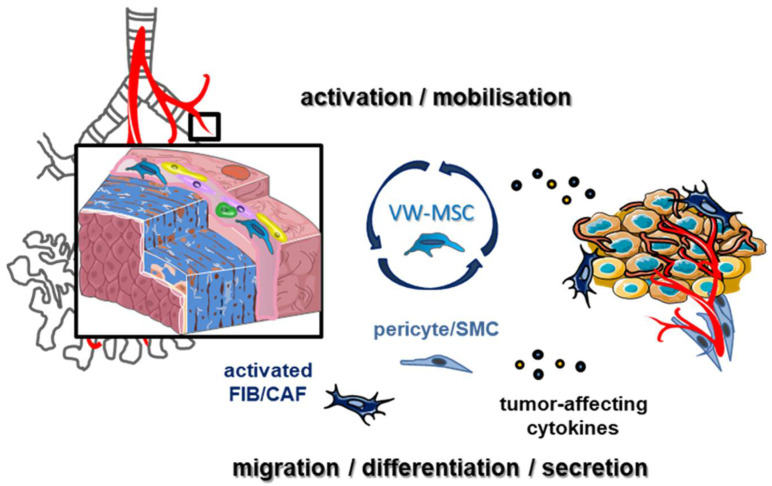
Lung-resident mesenchymal stem cells (LR-MSCs) potentially impact on lung cancer. LR-MSCs, predominately residing within the vasculogenic zone of adult lung arteries, are important orchestrators of normal lung homeostasis. Within this vascular stem cell niche also endothelial progenitor cells (displayed as yellow cells) and hematopoietic stem cells (green cells) reside. Following a pathological trigger, LR-MSCs seem not to appear sufficient to prevent lung tissue destructions and subsequent remodeling processes. LR-MSCs might at least partially contribute to lung diseases, particularly to lung cancer. Adjacent tumor cells (brown cells), tumor-derived factors, as well as associated hypoxia and inflammation might induce LR-MSCs mobilization and/or activation. Cancer-educated LR-MSCs in term contribute to the formation of malignant microenvironments fostering cancer progression and potentially therapy resistance by (i) differentiating into pericytes and smooth muscle cells (SMC) and, thus, impacting on vascular remodeling of newly formed tumor blood vessels, (ii) acquiring an activated fibroblast (myofibroblast or cancer associated-fibroblast, CAF) phenotype, and (iii) an altered tumor-promoting LR-MSC secretory profile engaging bi-directional intercellular interactions within lung tumors. Finally, although preclinical evidence is lacking up to now, LR-MSCs could be a cell of origin for lung cancer development or initiation. A detailed understanding of how LR-MSCs are implicated within the tumor stroma, and, thus, critically contribute to the heterogeneity and complexity of lung cancer biology, could foster strategies to manipulate these cells on site, finally paving an additional way for the discovery of potential drug targets for NSCLC patients.

**Figure 2 cancers-13-04637-f002:**
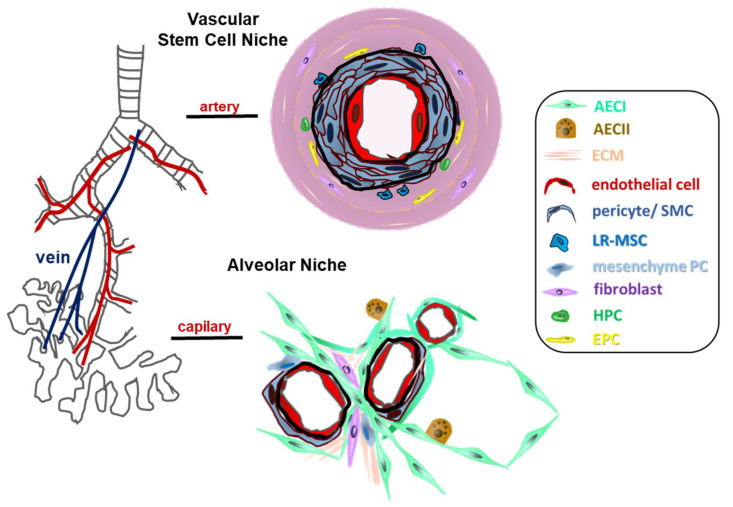
Mesenchymal (stem) cell niches within lungs. The vascular system is hierarchically organized and, thus, is composed of large, medium-sized, and the smallest blood vessels. Basically, a three-layered structure can be named: the innermost intima facing the vessel lumen, the middle media (tunica media) where vascular smooth muscle cells are oriented in a circle around the vascular lumen and form numerous layers (depending on vessel type, size, and location in the body), and the outermost adventitia. The latter one representing the interface between vessel wall and surrounding tissue, namely the perivascular space(s) that vary in dimension according to the type of blood vessel [105,106]. In contrast to the adventitia, the perivascular space does not display the circular organization. Perivascular cells are in close contact with the adventitia, and the adventitia and periadventitial/perivascular cells function in concert [107]. Lung-resident mesenchymal stem cells (LR-MSCs; blue) were shown to be predominantly localized within the vascular wall of larger lung blood vessels, the adventitial stem cells niche. Within that vascular stem cell niche, other vascular wall-resident stem and progenitor cells can be found: endothelial progenitor cells (EPCs depicted as yellow cells) and hematopoietic stem cells (HSCs depicted as green cells). The alveolar (stem cell) niche represents a combination of the vascular niche with the airway niche. Different mesenchymal cells, their respective progenitors, as well as pericytes with similar MSC morphologies and shared expressions of classical MSC markers were known to reside here within the interstitium between epithelial and endothelial cells (mesenchyme progenitor cells (PC), depicted as light blue cells).

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
