# Peer review of "Lung-Resident Mesenchymal Stem Cell Fates within Lung Cancer"

_cancers, 2021, doi:10.3390/cancers13184637_

Round 1
Reviewer 1 Report
The authors provided sufficient and broad insight into the literature of lung-resident mesenchymal stem cells and their role in cancer development. The article is well written, properly organized and shows the importance of this field.
I have only one minor comment. The authors stated that MSCs can differentiate into pericytes and these cells contribute to vascular remodelling and angio/vasculogenesis. Pericytes often considered as MSCs, and it has been shown that pericytes affect the adhesion, the migration and the proliferation of cancer cells and these mural cells presented a very prominent pro-metastatic effect as well (doi: 10.1002/1878-0261.12752.).
Taken together, the article is excellently written and should be considered for publication.
Author Response
First, we would like to thank for the positive evaluation of our manuscript enabling us the re-submission of our revised manuscript. According to the reviewer’s suggestions, we addressed and corrected the critical point. Changes were emphasized within the manuscript using yellow color.
We included an extra paragraph (and an extra Figure) explaining the difficulties in discriminating similar mesenchymal cells (including pericytes) in lungs. We also included the indicated and moreover important reference.
Reviewer 2 Report
The review manuscript titled “Lung-resident mesenchymal stem cell fates within lung cancer” by Sentek and Klein is a well-rounded and written review that introduces and covers in-depth the area of lung resident MSCs and proximal effects by cancer. This is one of the better reviews, I have had the pleasure to provide commentary on, it was nigh impossible to find anything to provide corrections that were not nitpicked. Sans any minor corrections this manuscript can be published in its current form.
Author Response
We would like to thank the reviewer very much for the positive evaluation of our manuscript and the endorsement for publication!
Reviewer 3 Report
This manuscript deals with the role of lung-resident mesenchymal stem cells (LR-MSCs) in regulating the lung cancer cells behaviour. Indeed, LR-MSC can be MSC from which can derive cancer associated fibroblasts (CAF) or MSC localized in association with vascular components of lung tumors.
From this review, one remain with the idea that LR-MSC are somehow an elusive kind of cell with several features similar to CAF or mural cells or fibroblast-like cells able to differentiate into other cell types. Perhaps, LR-MSC can also have some regulatory activity of immunity (although this point is not analysed in depth).
In other words, from this review I do not get any clear message to contextualize the LR-MSC. Perhaps, the addition of some items regarding phenotypic (markers characteristic of a given cell population) and functional features (cytokines, and other factors) specific (if they exist) for each of the populations described could help to understand the role of these cells.
Items on the interrelationship among the different cell populations can help again.
Perhaps, there is not enough information on the topic in the literature to distinguish among the cell types described. At least, MSC (mesenchymal stromal cells/mesenchymal stem cells) can become LR-MSC, CAF, mural cells and so on. Furthermore, how may I distinguish them when these cells are in the lung? By anatomic localization? If this is the case, a table or a cartoon should explain well this point.
Author Response
First, we would like to thank for the positive evaluation of our manuscript enabling us the re-submission of our revised manuscript. According to the reviewer’s suggestions, we addressed and corrected the critical point. Changes were emphasized within the manuscript using yellow color.
We included an extra paragraph (and an extra Figure) explaining the difficulties in discriminating similar mesenchymal cells in lungs. According to the reviewer’s suggestion we summarized the (limited) knowledge concerning the differential phenotypic markers of respective cells, their known differences concerning stem cell properties and their anatomic localization.
Round 2
Reviewer 3 Report
The authors have added a figure and a paragraph that can help the reader to understand the differences among LR-MSc and other stromal cells present in the lung.
I would say that the paper has been improved.